nanotechnology/materials science

zeolit, porous material, monolith, nanocrystal

**Author for correspondence:**
Yongfeng Li
e-mail: gdliyf@gdut.edu.cn

---

†These authors share first authorship based on equal contribution.

This article has been edited by the Royal Society of Chemistry, including the commissioning, peer review process and editorial aspects up to the point of acceptance.

# Self-standing zeolite foam monoliths with hierarchical micro–meso–macroporous structures

Jiawei Chen[1],[†], Fangfang Liu[1],[†], Yongfeng Li[1], Yongshen Dou[2], Sanmao Liu[2] and Liangjun Xiao[1]

[1]School of Chemical Engineering and Light Industry, Guangdong University of Technology, Guangzhou 510006, People's Republic of China
[2]Foshan Shunde Kinglei Environment and Technology Co. Ltd, Foshan 528308, People's Republic of China

YL, 0000-0003-0978-9274

The zeolite monoliths were synthesized by a facile polymer scaffold template assisted hydrothermal method. The selected foam-shaped template of a polyurethane (PU) foam monolith, was used to prepare the self-standing zeolite foam (ZF) monolithic materials. The obtained ZF products can preserve the same size, shape and macroporous network structure of the original PU foam scaffold template, although the zeolite nano-crystallites had been fully substituted for the PU template to form the new skeleton struts and walls. The as-synthesized ZF products demonstrated abundant hierarchical porosity (involving triple micro-, meso- and macropores). Meanwhile, compared with the conventional zeolite powders, the self-standing ZF monolithic materials exhibited greater total pore volume and nearly three times higher mesopore volume, suggesting wider applications as catalysts, catalyst supports and adsorbents in industry.

## 1. Introduction

Hierarchically structured porous materials with multiple porosities over lengths in the micro, meso to macro ranges could provide higher surface areas, higher pore volume ratios, higher accessibilities, ready mass transport properties and higher storage capacities compared to simple porous materials [1,2]. They are expected to be widely used in areas ranging from nanoscience to catalysis, separation, energy, life science and other industrial applications [3–6]. The macroporous polymers are usually used as scaffold templates for the preparation of hierarchically porous materials, around or within which the

precursor solution infiltrates and produces nanoparticles with the interparticulate micro-(or meso-)pores along with the macropores of the initial template [7–10]. Nowadays, the use of macroporous polymer gels, foams and films as templates has fabricated many monolithic hierarchical materials, such as silica foams [11–13], honeycomb-like mesoporous carbons [14], open-cell aerogel foams [15], silicoboron carbonitride foams [16] and monolith-type boron nitride foams [17].

The conventional zeolites are an important group of simple porous materials with large-scale applications in petroleum and environmental industry as heterogeneous catalysts and adsorbents, because of their unique properties such as high microporosity, high hydrothermal stability, molecular sieving behaviour and relatively economical production [18–20]. Owing to a technical limitation of pressure drop for the zeolites in the form of fine powders, the zeolite monoliths with hierarchical porosity have received a lot of attention recently, because of their low-pressure drop, excellent mass transfer and shape controllability [21–26]. Nowadays, the zeolite monoliths are usually prepared by depositing zeolite films on the surface of supports with tailored macropores [27–31]. However, low zeolite-to-support weight ratio and easy loss of zeolite films from support under repeated temperature swings restrict the improvement of available zeolite monoliths and their performance in the industry.

In order to overcome such practical problems, using macroporous polymer monoliths as scaffold templates to prepare new type zeolite monoliths might be a suitable choice. After removal of the polymer, the resulting product can keep the size, shape and structural properties of the original template to form self-standing zeolite monoliths with hierarchical porosity. In this study, we choose a polyurethane (PU) foam monolith as a polymer template to produce a self-standing ZSM-5 type zeolite foam (ZF) monolith, which has triple hierarchical micro–meso–macroporous structures.

## 2. Experimental

The PU foam template was taken from Yancheng Yiwei Daily Necessities Co. Ltd., China. TPAOH, TEOS and sodium aluminate were purchased from Shanghai Aladdin Bio-Chem Technology Co. Ltd., China. All chemicals were analytical reagents and used as received.

The ZSM-5 type ZF monolith was synthesized by a polymer scaffold template assisted hydrothermal method. First, a beaker containing 85 ml distilled water was placed on a magnetic stirrer, then 20 ml of 1.0 mol l$^{-1}$ TPAOH, 40 ml TEOS and 3.2 ml of 1.0 mol l$^{-1}$ sodium aluminate were added and dissolved in the water with a stirring speed of about 800 r.p.m. at room temperature. After 10 h continuous stirring, the zeolite precursor solution was obtained and transferred to a Teflon-lined autoclave. Then the yellow coloured cylinder-shape PU foam monolith was soaked in the zeolite precursor solution for 1 h. Subsequently, the autoclave was sealed and placed in a static oven for hydrothermal treatment at 120°C for different reaction times of 3–36 h. After the hydrothermal reaction, the solid sample was taken out, washed with distilled water and dried at 120°C for 3 h. The obtained pale yellow coloured sample was called the transition zeolite foam (noted as TZF-nh, n referring to the hydrothermal reaction time of 3–36 h). Finally, after calcination at 650°C with air flow for 6 h, the white coloured zeolite foam monolith (noted as ZF) product was obtained, which has the similar shape and size of the original PU foam template. As a comparison, the ZSM-5 type zeolite powder (noted as ZP) was also synthesized by the same hydrothermal method without adding the PU foam template.

Scanning electron microscopy (SEM) with energy dispersive spectrometer (EDS) was taken using a Thermo Fisher Phenom ProX Desktop electron microscope operated at 15 kV. Thermogravimetry (TG) curves were obtained by using a NETZSCH STA409PC system with the purge gas of dry air and a heating rate of 5 K min$^{-1}$. X-ray diffractometer (XRD) patterns were recorded on a Bruker D8 Advance diffractometer using Cu-K$\alpha$ radiation at 40 kV and 40 mA. Fourier transform infrared-attenuated total reflectance (ATR-FTIR) measurements were acquired on a Thermo Fisher Nicolet 6700 spectrometer equipped with a diamond crystal single bounce ATR attachment. Nitrogen adsorption/desorption isotherms were measured on an ASAP 2020 analyser.

## 3. Results and discussion

The used PU foam scaffold template was a yellow coloured and cylinder-shape monolith, as shown in the inset of figure 1a. The SEM image of the template in figure 1a further showed that the skeleton struts of the template were interconnected with each other to form many macropores, whose cross-sections were near circular with a diameter of 100–300 μm. Meanwhile, from the SEM images at high magnification, it was observed that the interior of the PU foam skeleton strut was solid (figure 1b), and the strut walls

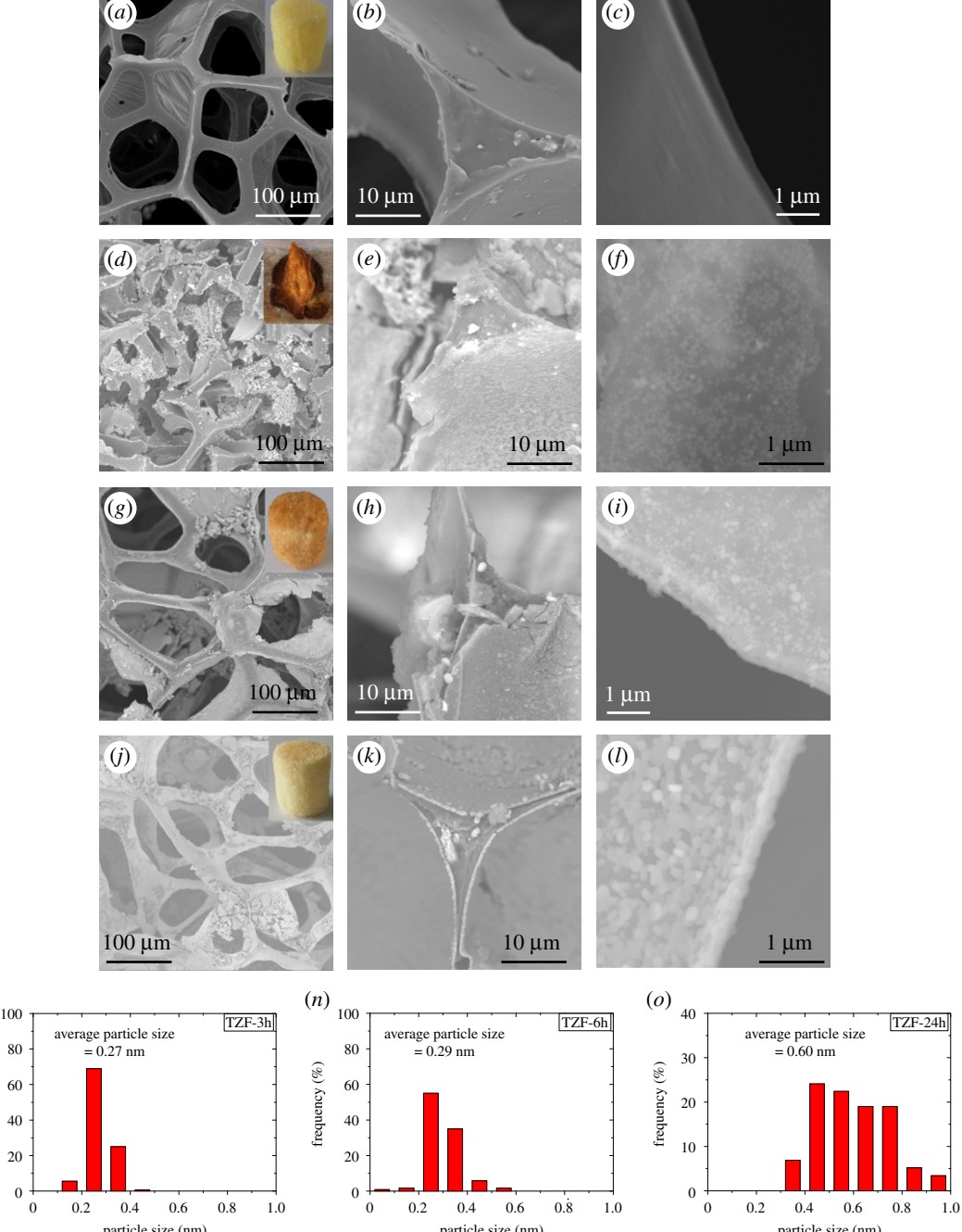

**Figure 1.** SEM and optical (inset) images of (*a–c*) pristine PU foam template, and TZF samples obtained with hydrothermal reaction times of (*d–f*) 3 h, (*g–i*) 6 h, and (*j–l*) 24 h. (*m–o*) Particle size distribution of the zeolite crystallites on the resultant TZF samples with hydrothermal reaction times of (*m*) 3 h, (*n*) 6 h, and (*o*) 24 h. Specifically, SEM images of (*a,d,g,j*) skeleton structure, (*b,e,h,k*) triangular cross-section of the strut, and (*c,f,i,l*) surface morphology of the strut wall.

were about 1 µm thick and very smooth on the surface (figure 1*c*). The results suggest that the pristine PU foam monolith template only has macroporous network structure.

The hydrolysis degradation behaviours of the PU template in the base aqueous solution mainly involves the band cleavage of organic components with water to form new substances [32,33]. Initially, the base-induced degradation of the PU template occurs in the hard segment by dissociating the relatively weak intermolecular hydrogen bonding between the urethane or ether groups of the adjacent PU chains [34]. Then the degradation further induces the hydrolysis of the C=O band of the urethane or ester group in the soft segment domain [35]. Finally, the segmentation of the PU chains ends up with the deterioration of the skeleton struts and walls of the PU template. That is to say, the hydrolysis time is one of the key factors for the decomposition of the PU foam scaffold template. Hence, while the PU foam template is

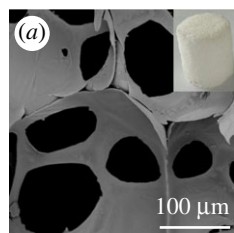 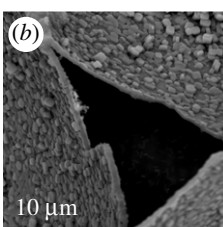 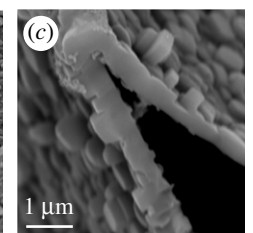 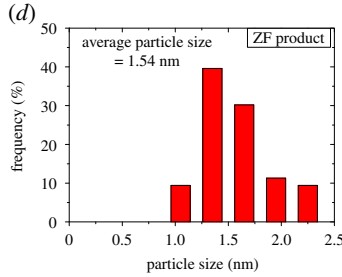

**Figure 2.** SEM and optical (inset) images of (*a*) skeleton structure, (*b*) triangular cross-section of the strut, (*c*) surface morphology of the strut wall, and (*d*) particle size distribution of the zeolite crystallites for the final ZF product after calcination at 650° with air flow for 6 h.

soaked in the zeolite precursor solution at 120°C, the hydrothermal reaction time plays a key role in the decomposition of the original PU scaffold template and the generation of new zeolite crystallites. While the hydrothermal reaction time was as short as 3 h, it was observed that the skeleton struts in the obtained TZF-3 h sample were mostly fractured (figure 1*d*), indicating that the sample cannot sustain the original cylinder shape of PU foam template (inset of figure 1*d*). From the SEM images of the TZF-3 h sample at high magnifications in figure 1*e* and *f*, it is found that the strut walls of the PU template had been partly replaced by zeolite crystallites, but the interior of skeleton struts still kept solid. Because the original PU skeleton walls were just partly replaced by zeolite crystallites, the boundaries between new-generated zeolite crystallites and the pristine PU template were loosely connected. Around such boundaries, the skeleton struts and walls were easily broken, resulting in the skeleton collapse of the TZF-3 h sample. Prolonging the hydrothermal reaction time to 6 h, the zeolite crystallites had been fully substituted for the PU skeleton walls (figure 1*i*). Hence, the obtained TZF-6 h sample can basically keep the similar cylinder shape and macroporous network of the pristine PU foam scaffold template (figure 1*g*). However, because the hydrothermal reaction time was still too short, the zeolite crystallites were rather small with an average size of 0.29 nm (figure 1*n*), and not closely arranged with each other, leading to some cracks on the surface of new-generated zeolite strut walls (figure 1*h*). While the hydrothermal reaction time was as long as over 24 h, a pale yellow coloured cylinder-shaped monolith of TZF-24 h sample was obtained, which had the same size and shape of the pristine PU foam template, as shown in the inset of figure 1*j*. The TZF-24 h sample also showed the nearly same macroporous network structure with the original PU template, besides some PU fragments left on the surface of zeolite skeleton walls (figure 1*j*). The SEM images at high magnifications of the TZF-24 h sample demonstrated that the zeolite crystallites had not only completely replaced the original strut walls of PU scaffold template, but also been closely aligned to form new uniform zeolite walls (figure 1*k*). The average size of the zeolite crystallites in TZF-24 h sample was 0.60 nm (figure 1*o*), and the thickness of the resultant zeolite walls was about 1 µm (figure 1*l*), in accordance with the thickness of the original PU template. That is to say, after hydrothermal reaction over 24 h, the obtained TZF samples can preserve the same size, shape and macroporous network structure of the original PU scaffold template. However, there were still some PU fragments on the surface of zeolite walls (figure 1*j*) and PU residues inside the zeolite struts (figure 1*k*).

The leftover PU fragments or residues in the TZF samples can be completely removed by calcination treatment at 650°C with air flow for 6 h, as shown in figure 2. Compared to the pale yellow colour of the TZF sample, the bright white colour of the ZF product indicates that the PU fragments in the TZF sample have been removed after calcination treatment because the original PU template is a yellow colour (inset of figure 2*a*). The SEM images of the ZF product further proved that the left PU fragments on the surface of zeolite walls and the PU residues inside the zeolite struts had been fully removed (figure 2*a* and *b*), leading to the new-generated zeolite skeleton struts which are hollow. The thickness of zeolite skeleton walls for the ZF product also remained about 1 µm (figure 2*c*), in line with the thickness of the original PU scaffold template, indicating that the new-generated zeolite walls are formed by completely substituting for original PU skeleton walls. However, the average particle size of zeolite crystallites in the ZF product increased to about 1.54 nm because of the high temperature treatment (figure 2*d*), and these zeolite nano-crystallites with coffin shapes were closely packed together to form new zeolite skeleton walls, which were rough and porous in contrast to original PU template surface. Therefore, the results reveal that the self-standing ZF monoliths can really be synthesized by the PU foam scaffold template assisted hydrothermal method, and the as-synthesized ZF product can maintain the same shape, size and macroporous network structure of the original PU foam template, although the whole skeleton struts and walls of original PU template have been fully replaced by new-generated zeolite nano-crystallites.

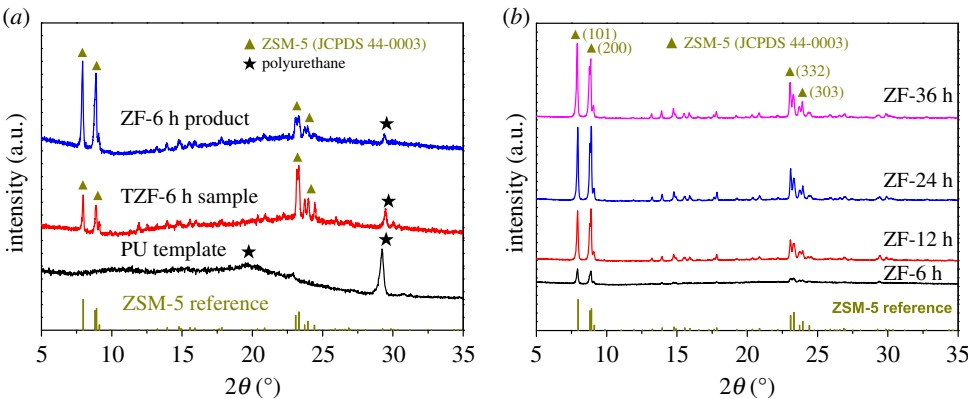

**Figure 3.** (a) XRD patterns of pristine PU foam template, TZF-6 h sample and final ZF-6 h product; (b) XRD patterns of final ZF products obtained with different hydrothermal reaction times.

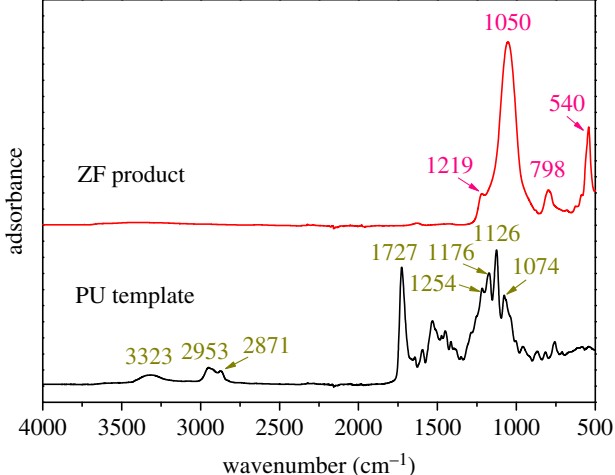

**Figure 4.** ATR-FTIR spectra of pristine PU foam template and final ZF product.

The XRD patterns of the TZF sample and final ZF product with the same hydrothermal reaction time of 6 h were compared in figure 3a. The XRD patterns of the pristine PU template and ZSM-5 reference are also shown in figure 3a. It is found that the PU template only had a broad diffraction peak at $2\theta = 19.6°$ and a sharp peak at $2\theta = 29.2°$, which is ascribed to the crystalline structure of PU [36]. After hydrothermal reaction for 6 h, the obtained TZF-6 h sample showed that the intensity of PU diffraction peaks decreased mark, and the new specific diffraction peaks might be assigned to the MFI structure of ZSM-5 [37], evidenced by the nearly same diffraction peaks with ZSM-5 reference (JCPDS 44-0003). Further, after calcination treatment, the final ZF product exhibited much stronger intensity of ZSM-5 diffraction peaks and little PU diffraction peaks. The results not only prove that the new-generated zeolite nano-crystallites are ZSM-5 species, but also confirm, in accordance with the SEM results, that the ZSM-5 zeolite crystallites in the ZF product have been fully substituted for the original PU scaffold template.

The XRD patterns of the ZF products obtained with different hydrothermal reaction times are presented in figure 3b. Compared to ZSM-5 reference (JCPDS 44-0003), all the ZF products had the similar diffraction peaks ascribed to ZSM-5 species, indicating that the hydrothermal reaction time has no effect on the crystal structure of ZSM-5 species in ZF products. Moreover, It is seen that the intensity of ZSM-5 diffraction peaks increased a lot with the increase of hydrothermal reaction time from 6 to 24 h, and remained nearly invariable in case of the hydrothermal reaction time over 24 h. This behaviour suggests that longer hydrothermal reaction times are favourable to obtain ZF products with higher degree of ZSM-5 crystallinity, and the suitable hydrothermal reaction time should be over 24 h.

The ATR-FTIR spectra of the pristine PU foam template and ZF product are presented in figure 4, and the IR band positions and corresponding group assignments for urethane and zeolite species are shown in table 1. The original PU template shows a series of characteristic adsorption bands at 3323, 1953, 2871, 1727, 1254, 1176, 1126 and 1074 cm$^{-1}$ in the ATR-FTIR spectrum, which are all assigned to PU species. In detail, the band around 3323 cm$^{-1}$ was related to N–H stretching vibration of urethane group

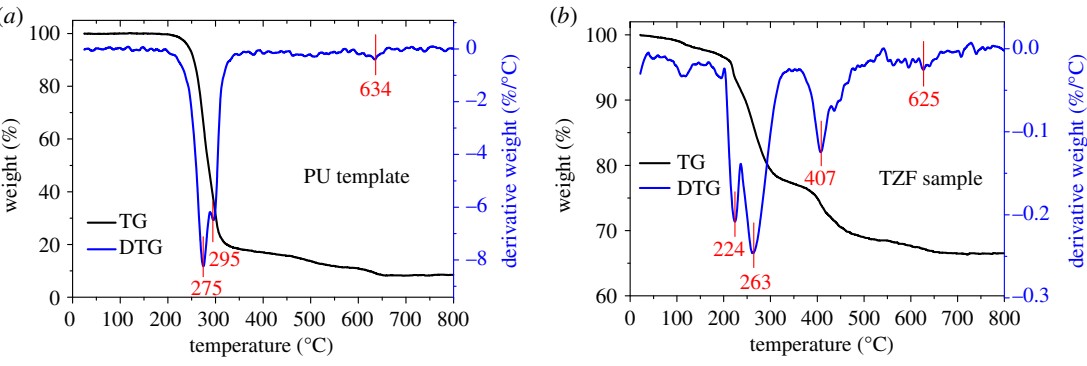

**Figure 5.** TG-DTG profiles of (*a*) pristine PU foam template and (*b*) the TZF sample.

**Table 1.** IR band positions and their corresponding group assignments for urethane and zeolite species.

| species | band position (cm$^{-1}$) | group assignment |
| --- | --- | --- |
| urethane | 3323 | stretching vibration of N–H in –NHCOO– |
| | 2953, 2871 | asymmetric/symmetric stretching vibration of C–H in –CH$_2$– group |
| | 1727 | free urethane C=O in urethane linkage |
| | 1254 | C–O–C in hard and soft segments of urethane |
| | 1176 | C–O–C in ester group |
| | 1126 | C–O–C in ether group |
| | 1074 | C–O–C in hard segments of urethane |
| zeolite | 1219, 1050 | asymmetric/symmetric stretching vibration of Si–O–Si in zeolite |
| | 798 | stretching vibration of Si–OH |
| | 540 | five-rings of T–O–T (T=Si or Al) structures in microporous zeolites |

(–NHCOO–) [38]; the bands at 2953 and 2871 cm$^{-1}$ corresponded to C–H stretching vibration, including asymmetric and symmetric stretching modes of methylene group [39]; the sharp band at 1727 cm$^{-1}$ was assigned to free urethane C=O in urethane linkage [40]; and the bands observed around 1254, 1176, 1126 and 1074 cm$^{-1}$ were known as the characteristic C–O–C stretching vibration band in ester and ether groups, and in hard and soft segment stretch of urethane [40]. Moreover, for the ZF product, it was observed from the ATR-FTIR spectrum in figure 4 that all the absorption bands characteristic of pristine PU template had disappeared, but four new bands at 540, 798, 1050 and 1219 cm$^{-1}$ were appeared, which are ascribed to zeolite structure. In details, the bands at 1219 and 1050 cm$^{-1}$ corresponded to the symmetric and asymmetric stretching vibration of Si–O–Si in zeolite; the band at 798 cm$^{-1}$ was related to stretching vibration of Si–OH; the sharp band at 540 cm$^{-1}$ was assigned to five-rings of T–O–T (T=Si or Al) in zeolite skeleton structures [41]. The results further certify that the original skeleton structure of PU template has been completely replaced by the new-generated zeolite skeleton structure in the ZF product obtained by the polymer scaffold template assisted hydrothermal method.

The TG-derivative TG (DTG) profiles were also used to analyse the calcination treatment process for the PU template and TZS sample, as shown in figure 5. For the pristine PU foam template, it is found that the thermal decomposition proceeds in three stages (figure 5*a*). The first stage occurred at a temperature range of 200–288°C with a weight loss of 54%, which is ascribed to the evaporation of capillary water and some volatile small molecules in PU foam [39]. The second stage of 288–344°C with a weight loss of 28% is conjectured as the thermal decomposition of N–H, C=O and C–O–C in urethane groups, urea linkages or ether groups, referring to the literature [40]. The third stage (greater than 500°C with a small weight loss of 5%) is mainly owing to the thermal degradation of hardly degradable composites in PU foam or segments generated in the second stage [36]. By contrast to the pristine PU template, it was found in figure 5*b* that the obtained TZF sample also had the three thermal decomposition stages of PU template at similar temperature ranges, but the decomposition weight losses decreased markedly and the decomposition temperature ranges shifted to lower temperatures. The result again proves that a

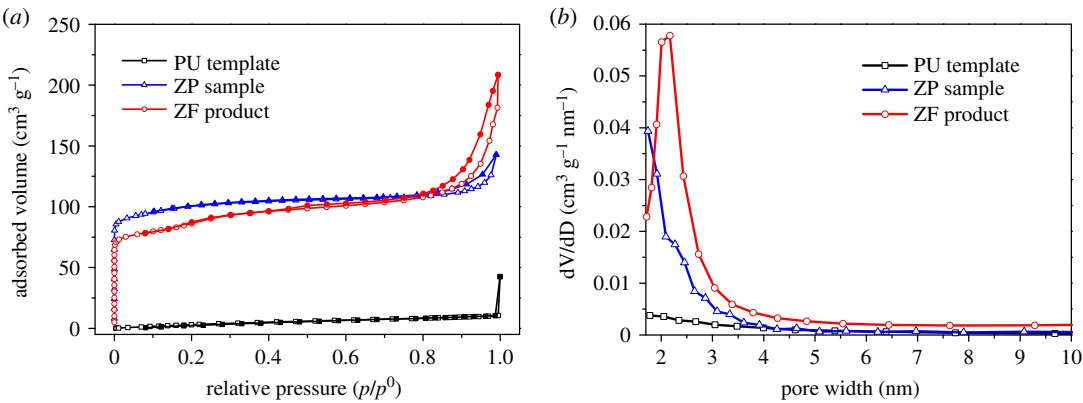

**Figure 6.** (a) Nitrogen adsorption/desorption isotherm plots (open symbol is for adsorption and closed for desorption points) and (b) BJH pore size distribution curves by adsorption branch of the isotherms of different materials.

few PU residues still existed in the TZF sample just after the hydrothermal reaction, but these PU residues were easily removed by simple thermal treatment. Meanwhile, a new weight loss at 362–476°C appeared in the TZF sample (figure 5b), which is assigned to the thermal decomposition of structure-forming $TPA^+$ ions residing within the zeolite channels [42]. Therefore, the TG-DTG results indicate that the suitable calcination temperature for TZF samples should be over 650°C, in order to completely remove the PU residues and TPAOH structure template agent.

By contrast to the smooth surface of the skeleton strut of the walls of original PU template, the new-generated zeolite skeleton walls of ZF products were formed by the aggregation of coffin-shaped ZSM-5 nano-crystallites and seemed rough (figure 2b), indicating that some meso or micropores may be generated among the packed zeolite crystallites. Meanwhile, the new skeleton struts of the ZF product were of hollow structure, which may also generate extra meso or macropores between zeolite strut walls. Most of the micropores can also be introduced by the ZSM-5 zeolite itself. In order to evaluate the hierarchical porosity of the obtained ZF product, its $N_2$ adsorption/desorption isotherm was measured, as shown in figure 6a. As a comparison, the $N_2$ adsorption/desorption isotherms of the pristine PU foam template and the conventional ZSM-5 ZP sample are also presented in figure 6a.

As shown in figure 6a, the isotherm of the pristine PU foam template was similar to type III shape, indicating that it is essentially nonporous or macroporous in nature. This is consistent with the SEM result of only the macroporous network on the smooth wall surface of the PU foam template. However, both zeolite materials of the ZP sample and ZF product exhibited a mixed type of I and IV isotherms with a steep uptake at very low $p/p^0$ resulting from micropore filling and a hysteresis loop at high $p/p$ [0] caused by mesopore capillary condensation [43]. This behaviour suggests the presence of meso-porosity with well-developed microporosity in the two materials. The micropores in both zeolite materials are mostly ascribed to the skeleton structure of the ZSM-5 zeolite with a sinusoidal channel of $0.55 \times 0.51$ nm and straight elliptical channel of $0.56 \times 0.53$ nm [44], and the type H3 hysteresis loop in figure 6a indicates that the secondary slit-like mesoporous structure is also formed in both ZP and ZF [43]. A comparison in the shape of isotherms further shows that the ZF product has higher mesoporous adsorption properties than the ZP sample. In order to explore more about the differences in porosity of both zeolite materials, their pore size distributions were calculated by the Barret, Joyner and Halenda (BJH) method based on a modified Kelvin equation and presented in figure 6b. Meanwhile, the Brunauer–Emmett–Teller (BET) surface area, micropore, mesopore and total pore volumes were also calculated from the adsorption branch of the isotherms, as shown in table 2. It is found that the ZP sample only had a few mesopores with width of 2–3 nm, besides the large micropores ascribed to ZSM-5 powder itself (figure 6b). Hence, the ZP sample showed large micropore volume of $0.11 \, \text{cm}^3 \, \text{g}^{-1}$ and small mesopore volume of $0.09 \, \text{cm}^3 \, \text{g}^{-1}$ (table 2). As a comparison to the ZP sample, the as-obtained ZF product exhibited a prominent peak at the mesopore size of 2.2 nm and a wider mesopore size range of 2–4 nm (figure 6b), and the ZF product also showed nearly three times higher mesopore volume and half lower micropore volume than the ZP sample, as shown in table 2. The results suggest that the polymer scaffold template assisted hydrothermal method is favourable to the formation of self-standing zeolite monolith materials with much higher mesopore volume than micropore volume. In compliance with the foregoing SEM analysis, the $N_2$ adsorption isotherm results also confirm that the as-synthesized self-standing ZF product indeed reveals many extra micro- and meso-pores except macroporous network from the PU foam

**Table 2.** Surface properties of different materials obtained from the nitrogen adsorption/desorption isotherms.

| sample | surface area[a] $(m^2\ g^{-1})$ | pore volume $(cm^3\ g^{-1})$ | | |
| --- | --- | --- | --- | --- |
| | | micropore[b] | mesopore[c] | total |
| PU template | 19 | — | — | — |
| ZP sample | 356 | 0.11 | 0.09 | 0.20 |
| ZF product | 282 | 0.06 | 0.24 | 0.30 |

[a]Calculated by the Brunauer–Emmett–Teller (BET) method.
[b]Calculated by the t-plot method.
[c]Calculated by the Barret–Joyner–Halenda (BJH) method.

scaffold template. That is to say, the final self-standing ZF monolith product has the triple hierarchical micro–meso–macroporous structures, and much higher mesopore volume than micropore volume.

## 4. Conclusion

In this work, it was shown that it is possible to synthesize the self-standing zeolite monolith with hierarchical porosity by a facile polymer scaffold template assisted hydrothermal method. The as-synthesized ZF monolith products can keep the same size, shape and macroporous network structure of the original scaffold template of a PU foam monolith, although the skeleton struts and walls of the original PU template had been completely replaced by the new-generated ZSM-5 zeolite nano-crystallites in the ZF products. During such a substitution synthesis process, many extra mesopores were produced among the packed zeolite crystallites or between zeolite skeleton strut walls. Considering the micropores introduced by the ZSM-5 zeolite itself and the macropores along with the original PU foam scaffold template, the as-synthesized self-standing ZF monolith can be regarded as having a triple hierarchical micro–meso–macroporous structures, and much higher mesopore volume than micropore volume. In contrast to the conventional ZPs, we believe that such self-standing ZF monolithic materials will find wider applications in petroleum and environmental industry as heterogeneous catalysts and adsorbents.

Data accessibility. Our data are available as part of the electronic supplementary material.
Authors' contributions. F. L. and J.C. carried out the laboratory work, participated in data analysis and drafted the manuscript; Y.L. conceived of the study, designed the study, carried out the data analysis and critically revised the manuscript; Y.D. and S.L. cooperated to collect and analyse the data; L.X. coordinated the laboratory work and helped draft the manuscript. All authors gave final approval for publication.
Competing interests. We declare we have no competing interests.
Funding. This work was financially supported by the National Natural Science Foundation of China (grant nos. 51678160, 21606051, 21576054), the Guangdong Province Science and Technology Project (grant no. 2016A020221033), the Guangzhou Science and Technology Project (grant no. 201704020202) and the Research Fund for Applied Science and Technology of Guangdong (grant no. 2016B020241003).

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
