## [Reviewer comments · Royal Society Open Science]

Review History

RSOS-200981.R0 (Original submission)

Review form: Reviewer 1

Is the manuscript scientifically sound in its present form?

Yes

Are the interpretations and conclusions justified by the results?

Yes

Is the language acceptable?

Yes

Do you have any ethical concerns with this paper?

No

Have you any concerns about statistical analyses in this paper?

No

Recommendation?

Accept with minor revision (please list in comments)

Comments to the Author(s)

The manuscript reported a simple synthesis method of the self-standing zeolite foam monolithic materials by a facile polymer scaffold template assisted by hydrothermal method. The as-prepared zeolite foam can not only preserve the size, shape and macropore structure of the pristine polymer foam template, but also generate abundant micro and mesopores. The triple pore structure and high mesopore volume of the obtained zeolite monoliths are attractive. The synthesized material is comprehensively characterized by SEM, XRD, IR, TG-DTG, BET and BJH. The manuscript is also well organized.

I only have one small concern about the process of decomposition of original PU template during hydrothermal reaction. The author mentioned this process but didn't provide much details about the reaction. It will be better if the authors can add more details about it.

This manuscript can make some revisions about the above concern for acceptance.

Review form: Reviewer 2

Is the manuscript scientifically sound in its present form?

No

Are the interpretations and conclusions justified by the results?

No

Is the language acceptable?

No

Do you have any ethical concerns with this paper?

No

Have you any concerns about statistical analyses in this paper?

Yes

Recommendation?

Major revision is needed (please make suggestions in comments)

Comments to the Author(s)

Review on manuscript "Self-standing zeolite foam monoliths with hierarchical micro-meso-macroporous structures" by Chen et al.

The manuscript describes production of zeolite monoliths by growing ZMS-5 crystals that have replaced the Polyurethane shape during the synthesis, thus producing a hierarchy of pores ranging from nano- to macro-pores. While the premise of the work is quite intriguing, and I really enjoyed seeing the data, the manuscript in its current form is not ready of publication, both because of the a) lack of detailed analysis on the available data and b) very-poor language used.

a) Lack of analysis. Authors present plenty of really nice data to show that the method of growing the zeolite monoliths works quite well. We have here: PDRD data on phase identification, SEM data, IF data, BET and TG data. As far as I can see only SEM data has been carefully looked at, the rest of the data was just ploncked in without really much analysis. For example, if you look at PXRD data on ZMS-5 zeolites and compare to your samples, you might learn more about their structure. Which peaks overlap and which do not? This is not clear at the moment from the figures and descriptions. How can you be sure that you have ZSM-5 zeolite just from 4 peaks?

IR data: give us a table of the resonances, how do they compare to literature, etc. TG data - did you do mass spec? You talk about what species are being removed during heating, do you have mass spec data to back it up?

SEM data is lovely, but why not do more analysis on the size distribution of zeolite crystallites? Etc. etc. Etc. Very disappointing to see such lack of detailed analysis.

b) Language! Currently, I can understand the content of the manuscript, but it is not written well. There are quite a number of grammatical errors and very strange phrases. There is a clear lack of clarity in many sections. All of those could be addressed, please do it!

Some comments:

In Experimental:

Line 5: You say vigorous, how vigorous, what RPM did you use?

What purity chemicals did you use?

Line 9: You mention that you soaked the PU form in the precursor solution, was it then removed and the precursor solution heated in the oven, or was the PU form and the precursor solution heated in the oven? OK, from the text below I can see that the PU form was in the oven, but please amend this sentence, to make it abundantly clear what sequence of steps was taken.

Results and discussion:

Line 34: "struts were staffed"??? I am not sure "stuffed" is an appropriate word to use here.

Figure 2 is very good. You can also use Figure 2b and get the size and the distribution of the crystallites, for example using the program JEdit or similar software.

Figures 3. Please redo this figure. The lines should be thinner and on figure 3b, please include ZSM-5 reference sample from literature. There may not be a complete match between your samples the reference ZSM-5, which is all right and could be explained. It will be good to see exactly which peaks are matching and which are not, this will give you more information about the crystallites that you have grown.

Figure 4. Same here, please make the lines thinner and give us in the table all of the peaks that you are seeing in IR. Between 1000 and 1200 cm^{-1} there will be an overlap of the peaks, but a careful assignment will be quite useful. Also, please compare those results to the literature on ZSM-5, that is quite widely available.

There are a lot of sentences in Results and Discussion, which I would like to change, but I will let you work on those. It almost feels as if the Introduction was written by a different person, with slightly better English. It feels as the results and discussion section was translated using a not very good translator.

Decision letter (RSOS-200981.R0)

Dear Dr LI:

Title: Self-standing zeolite foam monoliths with hierarchical micro-meso-macroporous structures
Manuscript ID: RSOS-200981

The editor assigned to your manuscript has now received comments from reviewers. We would like you to revise your paper in accordance with the referee and Subject Editor suggestions which can be found below (not including confidential reports to the Editor). Please note this decision does not guarantee eventual acceptance.

Please submit your revised paper before 25-Jul-2020. Please note that the revision deadline will expire at 00.00am on this date. If we do not hear from you within this time then it will be

assumed that the paper has been withdrawn. In exceptional circumstances, extensions may be possible if agreed with the Editorial Office in advance. We do not allow multiple rounds of revision so we urge you to make every effort to fully address all of the comments at this stage. If deemed necessary by the Editors, your manuscript will be sent back to one or more of the original reviewers for assessment. If the original reviewers are not available we may invite new reviewers.

On behalf of the Subject Editor Professor Anthony Stace and the Associate Editor Professor Kim Jelfs.

RSC Associate Editor:
Comments to the Author:
Please address seriously the reviewers' comments.

RSC Subject Editor:
Comments to the Author:
(There are no comments.)

Reviewers' Comments to Author:
Reviewer: 1

Comments to the Author(s)
The manuscript reported a simple synthesis method of the self-standing zeolite foam monolithic materials by a facile polymer scaffold template assisted by hydrothermal method. The as-prepared zeolite foam can not only preserve the size, shape and macropore structure of the pristine polymer foam template, but also generate abundant micro and mesopores. The triple pore structure and high mesopore volume of the obtained zeolite monoliths are attractive. The

synthesized material is comprehensively characterized by SEM, XRD, IR, TG-DTG, BET and BJH. The manuscript is also well organized.

I only have one small concern about the process of decomposition of original PU template during hydrothermal reaction. The author mentioned this process but didn't provide much details about the reaction. It will be better if the authors can add more details about it.

This manuscript can make some revisions about the above concern for acceptance.

Reviewer: 2

Comments to the Author(s)

Review on manuscript "Self-standing zeolite foam monoliths with hierarchical micro-meso-macroporous structures" by Chen et al.

The manuscript describes production of zeolite monoliths by growing ZMS-5 crystals that have replaced the Polyurethane shape during the synthesis, thus producing a hierarchy of pores ranging from nano- to macro-pores. While the premise of the work is quite intriguing, and I really enjoyed seeing the data, the manuscript in its current form is not ready of publication, both because of the a) lack of detailed analysis on the available data and b) very-poor language used.

a) Lack of analysis. Authors present plenty of really nice data to show that the method of growing the zeolite monoliths works quite well. We have here: PDRD data on phase identification, SEM data, IF data, BET and TG data. As far as I can see only SEM data has been carefully looked at, the rest of the data was just plonked in without really much analysis. For example, if you look at PXRD data on ZMS-5 zeolites and compare to your samples, you might learn more about their structure. Which peaks overlap and which do not? This is not clear at the moment from the figures and descriptions. How can you be sure that you have ZSM-5 zeolite just from 4 peaks? IR data: give us a table of the resonances, how do they compare to literature, etc. TG data - did you do mass spec? You talk about what species are being removed during heating, do you have mass spec data to back it up?

SEM data is lovely, but why not do more analysis on the size distribution of zeolite crystallites? Etc. etc. Etc. Very disappointing to see such lack of detailed analysis.

b) Language! Currently, I can understand the content of the manuscript, but it is not written well. There are quite a number of grammatical errors and very strange phrases. There is a clear lack of clarity in many sections. All of those could be addressed, please do it!

Some comments:

In Experimental:

Line 5: You say vigorous, how vigorous, what RPM did you use?

What purity chemicals did you use?

Line 9: You mention that you soaked the PU form in the precursor solution, was it then removed and the precursor solution heated in the over, or was the PU form and the precursor solution heated in the oven? OK, from the text below I can see that the PU form was in the oven, but please amend this sentence, to make it abundantly clear what sequence of steps was taken.

Results and discussion:

Line 34: "struts were staffed"??? I am not sure "stuffed" is an appropriate word to use here.

Figure 2 is very good. You can also use Figure 2b and get the size and the distribution of the crystallites, for example using the program JEdit or similar software.

Figures 3. Please redo this figure. The lines should be thinner and on figure 3b, please include ZSM-5 reference sample from literature. There may not be a complete match between your samples the reference ZSM-5, which is all right and could be explained. It will be good to see exactly which peaks are matching and which are not, this will give you more information about the crystallites that you have grown.

Figure 4. Same here, please make the lines thinner and give us in the table all of the peaks that you are seeing in IR. Between 1000 and 1200 cm^{-1} there will be an overlap of the peaks, but a careful assignment will be quite useful. Also, please compare those results to the literature on ZSM-5, that is quite widely available.

There are a lot of sentences in Results and Discussion, which I would like to change, but I will let you work on those. It almost feels as if the Introduction was written by a different person, with slightly better English. It feels as the results and discussion section was translated using a not very good translator.

Author's Response to Decision Letter for (RSOS-200981.R0)

See Appendix A.

RSOS-200981.R1 (Revision)

Review form: Reviewer 1

Is the manuscript scientifically sound in its present form?

Yes

Are the interpretations and conclusions justified by the results?

Yes

Is the language acceptable?

Yes

Do you have any ethical concerns with this paper?

No

Have you any concerns about statistical analyses in this paper?

No

Recommendation?

Accept as is

Comments to the Author(s)

Now it is good to be accepted.

Decision letter (RSOS-200981.R1)

Dear Dr LI:

Title: Self-standing zeolite foam monoliths with hierarchical micro-meso-macroporous structures
Manuscript ID: RSOS-200981.R1

It is a pleasure to accept your manuscript in its current form for publication in Royal Society Open Science. The chemistry content of Royal Society Open Science is published in collaboration with the Royal Society of Chemistry.

On behalf of the Subject Editor Professor Anthony Stace and the Associate Editor Professor Kim Jelfs.

RSC Associate Editor:
Comments to the Author:
(There are no comments.)

RSC Subject Editor:
Comments to the Author:
(There are no comments.)

Reviewer(s)' Comments to Author:
Reviewer: 1

Comments to the Author(s)
Now it is good to be accepted.

Appendix A

Dear Editors and Reviewers:

Thank you for your letter and for the reviewers' comments concerning our manuscript entitled "Self-standing zeolite foam monoliths with hierarchical micro-meso-macroporous structures" (Manuscript ID: RSOS-200981). Those comments are all valuable and very helpful for revising and improving our paper, as well as the important guiding significance to our researches. We have studied comments carefully and have made correction which we hope meet with approval. Revised portion are marked in red in the paper. The main corrections in the paper and the responds to the reviewer's comments are as follows:

Responds to the reviewer's comments:

Reviewer #1(Remarks to the Authors):

Question: The manuscript reported a simple synthesis method of the self-standing zeolite foam monolithic materials by a facile polymer scaffold template assisted by hydrothermal method. The as-prepared zeolite foam can not only preserve the size, shape and macropore structure of the pristine polymer foam template, but also generate abundant micro and mesopores. The triple pore structure and high mesopore volume of the obtained zeolite monoliths are attractive. The synthesized material is comprehensively characterized by SEM, XRD, IR, TG-DTG, BET and BJH. The manuscript is also well organized. I only have one small concern about the process of decomposition of original PU template during hydrothermal reaction. The author mentioned this process but didn't provide much details about the reaction. It will be better if the authors can add more details about it. This manuscript can make some revisions about the above concern for acceptance.

Response: Considering the reviewer's comment, we have added the details about the process of decomposition of the original PU template during hydrothermal reaction, as follows: "The hydrolysis degradation behaviors of PU template in base aqueous solution mainly involves the band cleavage of organic components with water to form new substances^{32, 33}. Initially, the base-induced degradation of the PU template occurs in the hard segment by dissociating the relatively weak intermolecular hydrogen bonding between the urethane or ether groups of the adjacent PU chains³⁴. Then the degradation further induces the hydrolysis of the C=O band of the urethane or ester group in the soft segment domain³⁵. Finally, the segmentation of the PU chains ends up with the deterioration of the skeleton struts and walls of the PU template. That is to

say, the hydrolysis time is one of the key factors for the decomposition of PU foam scaffold template.” in lines 41-53 (left column) on page 2 of the revised manuscript.

Reviewer #2(Remarks to the Authors):

Question: The manuscript describes production of zeolite monoliths by growing ZMS-5 crystals that have replaced the Polyurethane shape during the synthesis, thus producing a hierarchy of pores ranging from nano- to macro-pores. While the premise of the work is quite intriguing, and I really enjoyed seeing the data, the manuscript in its current form is not ready of publication, both because of the a) lack of detailed analysis on the available data and b) very-poor language used.

Response: Considering the reviewer's comment, the whole manuscript has been revised. Especially, the Results and Discussion part has been re-written. (a) For the comment of “lack of detailed analysis on the available data”, some figures related to SEM, XRD, ATR-IR, and Particle size distribution had been added or changed. And more analysis and discussion about the SEM, XRD, ATR-IR, TG-DTG, N₂ adsorption isotherm, and Particle size distribution data had also been added and amended in the revised manuscript. The detailed response and correction will be stated afterwards, aimed at each specific issue related to this reviewer's comment. (b) For the comment of “very-poor language used”, we have carefully revised the whole manuscript, especially the Results and Discussion part. Many grammatical errors and strange phrases have been corrected. And the problems about lack of clarity and coherence in many sections have also been emphasized to correct. The detailed correction and changes had been marked in red in the revised manuscript.

Question: a) Lack of analysis. Authors present plenty of really nice data to show that the method of growing the zeolite monoliths works quite well. We have here: PDRD data on phase identification, SEM data, IF data, BET and TG data. As far as I can see only SEM data has been carefully looked at, the rest of the data was just ploncked in without really much analysis. For example, if you look at PXRD data on ZMS-5 zeolites and compare to your samples, you might learn more about their structure. Which peaks overlap and which do not? This is not clear at the moment from the figures and descriptions. How can you be sure that you have ZSM-5 zeolite just from 4 peaks? IR data: give us a table of the resonances, how do they compare to literature, etc. TG data – did you do mass spec? You talk about what species are being removed during heating, do you have mass spec

data to back it up? SEM data is lovely, but why not do more analysis on the size distribution of zeolite crystallites? Etc. etc. Etc. Very disappointing to see such lack of detailed analysis.

Response: Considering the reviewer's comment, the figures related to SEM, XRD, ATR-IR, and Particle size distribution curve has been added or changed. And more detailed analysis and discussion about the SEM, XRD, ATR-IR, TG-DTG, N₂ adsorption isotherm, and Particle size distribution data have been also added in the manuscript. The details are as follows:

- (1) For the XRD data, the line of ZSM-5 reference (JCPDS 44-0003) from literature had been added in Figure 3. And from Figure 3a and b, it was found that our obtained ZP products had nearly same diffraction peaks with the ZSM-5 reference, proving that the new-generated zeolite crystallites in ZP products are ascribed to ZSM-5 species. The detailed descriptions and analysis were also added in lines 24-28 (left column) and 36-39 (left column) on page 3 of the revised manuscript.
- (2) For the IR data, a new table was added ---- “Table 1 IR band positions and their corresponding group assignments for urethane and zeolite species”. With the aid of this table, the main IR peaks of original PU scaffold template and final ZF product had been carefully assigned to different characteristic groups, referring to literatures. The detailed analysis and discussion were also added in lines 49-57 (left column) and 1-18 (right column) on page 3 of the revised manuscript.
- (3) For the TG data, we are very sorry for our ambiguous statement. We did not do the mass spectra, and we talked about what species are being removed during heating just referring to the literatures. Our used PU template sample shows the similar thermal decomposition stages with the report of literatures. So the results in literatures about what species are being removed during different thermal decomposition stages were used to back up our descriptions for the TG-DTG data of pristine PU template and TZF sample. The detailed analysis and discussion were also added in lines 24-49 (right column) on page 3 of the revised manuscript.
- (4) For the SEM data, we added three new bar graphs in Figure 1m, n and o to show the particle size and distribution of zeolite crystallites on the resultant TZF samples obtained with different hydrothermal reaction times, and a new bar graphs in Figure 2d to reveal the particle size and distribution of zeolite crystallites on final ZF product, using Nano Measurer 1.2 software. The related analysis and discussion were also added in lines 21-24, and 35-37 (right column)

on page 2, and 4-7 (left column) on page 3 of the revised manuscript.

(5) For N₂ adsorption isotherm data, we emphasized to amend the analysis and discussion about the difference in porosity of pristine PU template, conventional ZSM-5 powder and our ZSM-5 foam monolith samples, and the formation mechanism of mesopores in ZF product, as shown in lines 51-57 (right column) on page 3 and lines 1-5, 9-51 (left column) on page 4 of the revised manuscript.

Question: b) Language! Currently, I can understand the content of the manuscript, but it is not written well. There are quite a number of grammatical errors and very strange phrases. There is a clear lack of clarity in many sections. All of those could be addressed, please do it!

Response: Considering the reviewer's comment, the whole manuscript had been revised. Especially, the Results and Discussion part has been re-written. Many grammatical errors and strange phrases have been corrected. And the problems about lack of clarity and coherence in many sections have also been emphasized to correct. The detailed correction and changes had been marked in red in the revised manuscript.

Question: In Experimental: Line 5: You say vigorous, how vigorous, what RPM did you use? What purity chemicals did you use?

Response: Considering the reviewer's comment, we revised the sentence as follows: "...were added and dissolved in the water with a stirring speed of about 800 revolutions per minute (RPM) at room temperature." in lines 47-49 (right column) on page 1 of the revised manuscript.

Considering the reviewer's comment, we added the detailed illustration about the chemicals used in our experiments as follows: "The PU foam template was taken from Yancheng Yiwei Daily Necessities Co., Ltd., China. TPAOH, TEOS and sodium aluminate were purchased from Shanghai Aladdin Bio-Chem Technology Co., Ltd., China. All chemicals were analytical reagent and used as received." in lines 38-42 (right column) on page 1 of the revised manuscript.

Question: In Experimental: Line 9: You mention that you soaked the PU form in the precursor solution, was it then removed and the precursor solution heated in the oven, or was the PU form and the precursor solution heated in the oven? OK, from the text below I

can see that the PU form was in the oven, but please amend this sentence, to make it abundantly clear what sequence of steps was taken.

Response: We are very sorry for our ambiguous statement. Considering the reviewer's comment, we amended the sentence as follows: "Then the yellow coloured cylinder-shape PU foam monolith was soaked in the zeolite precursor solution for 1 h. Subsequently the autoclave was sealed and placed in a static oven for hydrothermal treatment at 120°C for different reaction time of 3-36 h." in lines 1-5 (left column) on page 2 of the revised manuscript.

Question: Results and discussion: Line 34: "struts were staffed"??? I am not sure "stuffed" is an appropriate word to use here.

Response: We are very sorry for our ambiguous statement. The word "staffed" was used to point out that the interior of the skeleton struts in the pristine PU template is solid. Considering the reviewer's comment, we changed the sentence as follows: "it was observed that the interior of PU foam skeleton strut was solid" in lines 36-37 (left column) on page 2 of the revised manuscript.

Question: Results and discussion: Figure 2 is very good. You can also use Figure 2b and get the size and the distribution of the crystallites, for example using the program JEdit or similar software.

Response: Considering the reviewer's comment, we added "particle size distribution curve of the zeolite crystallites in the ZF product" in original Figure 2, using Nano Measurer 1.2 software. And the related discussion about the size and the distribution of the zeolite crystallites was also added in lines 4-7 (left column) on page 3 of the revised manuscript.

Question: Results and discussion: Figures 3. Please redo this figure. The lines should be thinner and on figure 3b, please include ZSM-5 reference sample from literature. There may not be a complete match between your samples the reference ZSM-5, which is all right and could be explained. It will be good to see exactly which peaks are matching and which are not, this will give you more information about the crystallites that you have grown.

Response: Considering the reviewer's comment, all the curves in Figure 3 had been redo. The lines had been changed thinner, and the line of ZSM-5 reference (JCPDS

44-0003) from literature had also been added. And from Figure 3a and b, it was found that our obtained ZP products had nearly same diffraction peaks with the ZSM-5 reference, proving that the new-generated zeolite crystallites in ZP products are ascribed to ZSM-5 species. The detailed analysis and discussion were also added in lines 24-28 (left column) and 36-39 (left column) on page 3 of the revised manuscript.

Question: Results and discussion: Figure 4. Same here, please make the lines thinner and give us in the table all of the peaks that you are seeing in IR. Between 1000 and 1200 cm^{-1} there will be an overlap of the peaks, but a careful assignment will be quite useful. Also, please compare those results to the literature on ZSM-5, that is quite widely available.

Response: Considering the reviewer's comment, all the lines had been changed thinner. And a new table was added ---- "Table 1 IR band positions and their corresponding group assignments for urethane and zeolite species". With the aid of this table, the main IR peaks of original PU scaffold template and final ZF product had been carefully assigned to different characteristic groups, referring to literatures. The detailed analysis and discussion were also added in lines 49-57 (left column) and 1-18 (right column) on page 3 of the revised manuscript.

Question: There are a lot of sentences in Results and Discussion, which I would like to change, but I will let you work on those. It almost feels as if the Introduction was written by a different person, with slightly better English. It feels as the results and discussion section was translated using a not very good translator.

Response: Considering the reviewer's comment, we have carefully revised the whole manuscript, especially the Results and Discussion part. Many grammatical errors and strange phrases had been corrected. And the problems about lack of clarity and coherence in many sections had also been emphasized to correct. The detailed correction and changes had been marked in red in the revised manuscript.

We tried our best to improve the manuscript and made some changes in the revised manuscript. These changes will not influence the content and framework of the paper. And here we did not list the changes but marked in red in revised paper.

We appreciate for Editors/Reviewers' warm work earnestly, and hope that the

correction will meet with approval.

Once again, thank you very much for your comments and suggestions.

Yours sincerely,

Yongfeng Li